# WHAT CAN LEARNED INTRINSIC REWARDS CAPTURE?

## ABSTRACT

Reinforcement learning agents can include different components, such as policies, value functions, state representations, and environment models. Any or all of these can be the *loci* of knowledge, i.e., structures where knowledge, whether given or learned, can be deposited and reused. Regardless of its composition, the objective of an agent is behave so as to maximise the sum of a suitable scalar function of state: the *reward*. As far as the learning algorithm is concerned, these rewards are typically given and immutable. In this paper we instead consider the proposition that the reward function itself may be a good locus of knowledge. This is consistent with a common use, in the literature, of hand-designed intrinsic rewards to improve the learning dynamics of an agent. We adopt the multi-lifetime setting of the Optimal Rewards Framework, and propose to meta-learn an intrinsic reward function from experience that allows agents to maximise their extrinsic rewards accumulated until the end of their lifetimes. Rewards as a locus of knowledge provide guidance on "what" the agent should strive to do rather than on knowledge of "how" the agent should behave that is more directly captured in policies or value functions for example. Thus, our focus here is on demonstrating the following: (1) that it is feasible to meta-learn good reward functions, (2) to show that the learned reward functions can capture interesting kinds of "what" knowledge, and (3) that because of the indirectness of this form of knowledge the learned reward functions can generalise to other kinds of agents and to changes in the dynamics of the environment.

Reinforcement learning agents can store knowledge in their policies, value functions, state representations, and models of the environment dynamics. These components can be the *loci* of knowledge in the sense that they are structures in which knowledge, either learned from experience by the agent's algorithm or given by the agent-designer, can be deposited and reused. The objective of the agent is defined by a reward function, and the goal is to learn to act so as to optimise cumulative rewards. In this paper we consider the proposition that the reward function itself is a good locus of knowledge. This is unusual in that most prior work treats the reward as given and immutable, at least as far as the learning algorithm is concerned. At the same time, especially in challenging reinforcement-learning problems, agent designers do find it convenient to modify the reward function given to the agent to facilitate learning. It is therefore useful to distinguish between two kinds of reward functions (Singh et al., 2010): *extrinsic* rewards define the task and capture the designer's preferences over agent behaviour, whereas *intrinsic* rewards serve as helpful signals to improve the learning dynamics of the agent. Intrinsic rewards are typically hand-designed and then often added to the immutable extrinsic rewards to form the reward optimised by the agent.

Most existing work on intrinsic rewards falls into two broad categories: task-dependent and task-independent. Both are typically designed by hand. Hand-designing *task-dependent* rewards can be fraught with difficulty as even minor misalignment between the actual reward and the intended bias can lead to unintended and sometimes catastrophic consequences (Clark & Amodei, 2016). *Task-independent* intrinsic rewards are also typically hand-designed, often based on an intuitive understanding of animal/human behaviour or on heuristics on desired exploratory behaviour. It can, however, be hard to match such task-independent intrinsic rewards to the specific learning dynamics induced by the interaction between agent and environment. The motivation for this paper is our interest in the comparatively under-explored possibility of learned (not hand-designed) task-dependent intrinsic rewards (see Zheng et al., 2018, for previous work).

We emphasise that it is *not* our objective to show that rewards are a *better* locus of learned knowledge than others; the best locus likely depends on the kind of knowledge that is most useful in a given

task. In particular, knowledge captured in rewards provides guidance on "what" the agent should strive to do while knowledge captured in policies provides guidance on "how" an agent should behave. Knowledge about "what" captured in rewards is indirect and thus slower to make an impact on behaviour because it takes effect through learning, while knowledge about "how" can directly have an immediate impact on behaviour. At the same time, because of its indirectness the former can generalise better to changes in dynamics and learning architectures. Therefore, instead of comparing different loci of knowledge, the purpose of this paper is to show that it is feasible to capture useful learned knowledge in rewards and to study the kinds of knowledge that can be captured.

How should we measure the usefulness of a learned reward function? Ideally, we would like to measure the effect the learned reward function has on the learning dynamics. Of course, learning happens over multiple episodes, indeed it happens over an entire lifetime. Therefore, we choose *lifetime return*, the cumulative extrinsic reward obtained by the agent over its entire lifetime, as the main objective. To this end, we adopt the multi-lifetime setting of the Optimal Rewards Framework (Singh et al., 2009) in which an agent is initialised randomly at the start of each lifetime and then faces a stationary or non-stationary task drawn from some distribution. In this setting, the only knowledge that is transferred across lifetimes is the reward instead of the policy. Specifically, the goal is to learn a single intrinsic reward function that, when used to adapt the agent's policy using a standard episodic RL algorithm, ends up optimising the cumulative extrinsic reward over its lifetime.

In previous work, good reward functions were found via exhaustive search, limiting the range of applicability of the framework. Here, we develop a more scalable gradient-based method (Xu et al., 2018c) for learning the intrinsic rewards by exploiting the fact the interaction between the policy update and the reward function is differentiable (Zheng et al., 2018). Since it is infeasible to backpropgate through the full computation graph that spans across the entire lifetime, we truncate the unrolled computation graph of learning updates up to some horizon. However, we handle the long-term credit assignment by using a lifetime value function that estimates the remaining lifetime return, which needs to take into account changing policies. Our main *scientific* contributions are a sequence of empirical studies on carefully designed environments that show how our learned intrinsic rewards can capture interesting regularities in the interaction between a learning agent and an environment sampled from a distribution, and how the learned intrinsic reward can generalise to changed dynamics and agent architectures. Collectively, our contributions present an effective approach to the discovery of intrinsic rewards that can help an agent optimise the extrinsic rewards collected in a lifetime.

## 1 RELATED WORK

**Hand-designed Rewards** There is a long history of work on designing rewards to accelerate learning in reinforcement learning (RL). Reward shaping aims to design task-specific rewards towards known optimal behaviours, typically requiring domain knowledge. Both the benefits (Randlov & Alstrm, 1998; Ng et al., 1999; Harutyunyan et al., 2015) and the difficulty (Clark & Amodei, 2016) of task-specific reward shaping have been studied. On the other hand, many intrinsic rewards have been proposed to encourage exploration, inspired by animal behaviours. Examples include prediction error (Schmidhuber, 1991b; Gordon & Ahissar, 2011; Mirolli & Baldassarre, 2013; Pathak et al., 2017; Schmidhuber, 1991a), surprise (Itti & Baldi, 2006), weight change (Linke et al., 2019), and state-visitation counts (Sutton, 1990; Poupart et al., 2006; Strehl & Littman, 2008; Bellemare et al., 2016; Ostrovski et al., 2017). Although these kinds of intrinsic rewards are not domain-specific, they are often not well-aligned with the task that the agent tries to solve, and ignores the effect on the agent's learning dynamics. In contrast, our work aims to learn intrinsic rewards from data that take into account the agent's learning dynamics without requiring prior knowledge from a human.

**Rewards Learned from Data** There have been a few attempts to learn useful intrinsic rewards from data. The optimal reward framework (Singh et al., 2009) proposed to learn an optimal reward function that allows agents to solve a distribution of tasks quickly using random search. We revisit this problem in this paper and propose a more scalable gradient-based approach. Although there have been follow-up works (Sorg et al., 2010; Guo et al., 2016) that uses a gradient-based method, they consider a non-parameteric policy using Monte-Carlo Tree Search (MCTS). Our work is closely related to LIRPG (Zheng et al., 2018) which proposed a meta-gradient method to learn intrinsic rewards. However, LIRPG considers a single task in a single lifetime with a myopic episode return

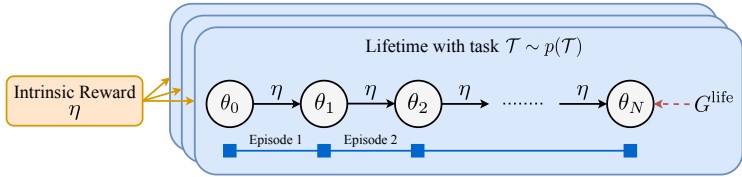

Figure 1: Illustration of the proposed intrinsic reward learning framework. The intrinsic reward $r_\eta$ is used to update the agent's parameter $\theta_i$ throughout its lifetime which consists of many episodes. The goal is to find the optimal intrinsic reward parameters $\eta^*$ across many lifetimes that maximises the lifetime return ($G^{\text{life}}$) given any randomly initialised agents and possibly non-stationary tasks drawn from some distribution $p(\mathcal{T})$.

objective, which is limited in that it does not allow exploration across episodes or generalisation to different agents.

**Meta-learning for Exploration** Meta-learning (Schmidhuber et al., 1996; Thrun & Pratt, 1998) has recently received considerable attention in RL. Recent advances include few-shot adaptation (Finn et al., 2017a), few-shot imitation (Finn et al., 2017b; Duan et al., 2017), model adaptation (Clavera et al., 2018), and inverse RL (Xu et al., 2018a). In particular, our work is closely related to the prior work on meta-learning good exploration strategies (Wang et al., 2016; Duan et al., 2016; Stadie et al., 2018; Xu et al., 2018b) in that both perform temporal credit assignment across episode boundaries by maximising rewards accumulated beyond an episode. Unlike the prior work that aims to learn an exploratory policy, our framework indirectly drives exploration via a reward function which can be reused by different learning agents as we show in this paper (Section 5.1).

**Meta-learning of Agent Update** There have been a few studies that directly meta-learn how to update the agent's parameters via meta-parameters including discount factor and returns (Xu et al., 2018c), auxiliary tasks (Schlegel et al., 2018; Veeriah et al., 2019), unsupervised learning rules (Metz et al., 2019), and RL objectives (Chebotar et al., 2019). Our work also belongs to this category in that our meta-parameters are the reward function used in the agent's update. In particular, our multi-lifetime formulation is similar to ML³ (Chebotar et al., 2019). However, we consider the long-term lifetime return as objective to perform cross-episode temporal credit assignment as opposed to the myopic episodic objective in ML³.

## 2 THE OPTIMAL REWARD PROBLEM

We first introduce some terminology.

- **Agent**: A learning system interacting with an environment. On each step $t$ the agent selects an action $a_t$ and receives from the environment an observation $s_{t+1}$ and an *extrinsic* reward $r_{t+1}$ defined by a task $\mathcal{T}$. The agent chooses actions based on a policy $\pi_\theta(a_t|s_t)$ parameterised by $\theta$.
- **Episode**: A finite sequence of agent-environment interactions until the end of the episode defined by the task. An episode return is defined as: $G^{\text{ep}} = \sum_{t=0}^{T_{\text{ep}}-1} \gamma^t r_{t+1}$, where $\gamma$ is a discount factor, and the random variable $T_{\text{ep}}$ gives the finite number of steps until the end of the episode.
- **Lifetime**: A finite sequence of agent-environment interactions until the end of training defined by an agent-designer, which can include multiple episodes. The *lifetime return* is $G^{\text{life}} = \sum_{t=0}^{T-1} \gamma^t r_{t+1}$, where $\gamma$ is a discount factor, and $T$ is the number of steps in the lifetime.
- **Intrinsic reward**: A reward function $r_\eta(\tau_{t+1})$ parameterised by $\eta$, where $\tau_t = (s_0, a_0, r_1, d_1, s_1, \ldots, r_t, d_t, s_t)$ is a lifetime history with (binary) episode terminations $d_i$.

The Optimal Reward Problem (Singh et al., 2010), illustrated in Figure 1, aims to learn the parameters of the intrinsic reward such that the resulting rewards achieve a learning dynamic for an RL agent that maximises the lifetime (extrinsic) return on tasks drawn from some distribution. Formally, the optimal reward function is defined as:

$$\eta^* = \arg\max_\eta J(\eta) = \arg\max_\eta \mathbb{E}_{\theta_0 \sim \Theta, \mathcal{T} \sim p(\mathcal{T})} \left[ \mathbb{E}_{\tau \sim p_\eta(\tau|\theta_0)} \left[ G^{\text{life}} \right] \right], \tag{1}$$

where $\Theta$ and $p(\mathcal{T})$ are an initial policy distribution and a distribution over possibly non-stationary tasks respectively, and $G^{\text{life}} = \sum_{t=0}^{T-1} \gamma^t r_{t+1}$ is a lifetime return. The likelihood of a lifetime history

---

**Algorithm 1** Learning intrinsic rewards across multiple lifetimes via meta-gradient

---

**Input**: $p(\mathcal{T})$: Task distribution, $\Theta$: Randomly-initialised policy distribution
Initialise intrinsic reward function $\eta$ and lifetime value function $\phi$
**repeat**
    Initialise task $\mathcal{T} \sim p(\mathcal{T})$ and policy $\theta \sim \Theta$
    **while** lifetime not ended **do**
        $\theta_0 \leftarrow \theta$
        **for** $k = 1, 2, \ldots, N$ **do**
            Generate a trajectory using $\pi_{\theta_{k-1}}$
            Update policy $\theta_k \leftarrow \theta_{k-1} + \alpha \nabla_{\theta_{k-1}} J_\eta(\theta_{k-1})$ using intrinsic rewards $r_\eta$ (Eq. 2)
        **end for**
        Update intrinsic reward function $\eta$ using Eq. 3
        Update lifetime value function $\phi$ using Eq. 4
        $\theta \leftarrow \theta_N$
    **end while**
**until** $\eta$ converges

---

$\tau$ is $p_\eta(\tau|\theta_0) = p(s_0) \prod_{t=0}^{T-1} \pi_{\theta_t}(a_t|s_t) p(d_{t+1}, r_{t+1}, s_{t+1}|s_t, a_t)$, where $\theta_t = f(\theta_{t-1}, \eta)$ is a policy parameter as updated with update function $f$, which is policy gradient in this paper.[1] Note that the optimisation of $\eta$ spans multiple lifetimes, each of which can span multiple episodes.

Using the lifetime return $G^{\text{life}}$ as the objective instead of the conventional episodic return $G^{\text{ep}}$ allows exploration across multiple episodes as long as the lifetime return is maximised in the long run. In particular, when the lifetime is defined as a fixed number of episodes, we find that the lifetime return objective is sometimes more beneficial than the episodic return objective even in terms of the episodic return performance measure. However, different objectives (e.g., final episode return) can be considered depending on the definition of what a good reward function is.

## 3 META-LEARNING INTRINSIC REWARD

We propose a meta-gradient approach (Xu et al., 2018c; Zheng et al., 2018) to solve the optimal reward problem. At a high-level, we sample a new task $\mathcal{T}$ and a new random policy parameter $\theta$ at each lifetime iteration. We then simulate an agent's lifetime by updating the parameter $\theta$ using an intrinsic reward function $r_\eta$ (Section 3.1) with policy gradient (Section 3.2). In the meantime, we compute the meta-gradient by taking into account the effect of the intrinsic rewards on the policy parameters to update the intrinsic reward function with a lifetime value function (Section 3.3). Algorithm 1 gives an overview of our algorithm. The following sections describe the details.

### 3.1 INTRINSIC REWARD AND LIFETIME VALUE FUNCTION ARCHITECTURES

The intrinsic reward function is a recurrent neural network (RNN) parameterised by $\eta$, which produces a scalar reward on arriving in state $s_t$ by taking into account the history of an agent's lifetime $\tau_t = (s_0, a_0, r_1, d_1, s_1, ..., r_t, d_t, s_t)$. We claim that giving the lifetime history across episodes as input is crucial for balancing exploration and exploitation, for instance by capturing how frequently a certain state is visited to determine an exploration bonus reward. The lifetime value function is a separate recurrent neural network parameterised by $\phi$, which takes the same inputs as the intrinsic reward function and produces a scalar value estimation of the expected future return within the lifetime.

### 3.2 POLICY UPDATE ($\theta$)

Each agent interacts with an environment and a task sampled from a distribution $\mathcal{T} \sim p(\mathcal{T})$. However, instead of directly maximising the extrinsic rewards defined by the task, the agent maximises

---

[1]We assume that the policy parameter is updated after each time-step throughout the paper for brevity. However, the parameter can be updated less frequently in practice.

the intrinsic rewards ($r_\eta$) by using policy gradient (Williams, 1992; Sutton et al., 2000):

$$J_\eta(\theta) = \mathbb{E}_\theta\left[\sum_{t=0}^{T_{\text{ep}}-1} \bar\gamma^t r_\eta(\tau_{t+1})\right] \qquad \nabla_\theta J_\eta(\theta) = \mathbb{E}_\theta\left[G^{\text{ep}}_{\eta,t}\nabla_\theta \log \pi_\theta(a|s)\right], \qquad (2)$$

where $r_\eta(\tau_{t+1})$ is the intrinsic reward at time $t$, and $G^{\text{ep}}_{\eta,t} = \sum_{k=t}^{T_{\text{ep}}-1} \bar\gamma^{k-t} r_\eta(\tau_{k+1})$ is the return of the intrinsic rewards accumulated over an episode with discount factor $\bar\gamma$.

## 3.3 INTRINSIC REWARD ($\eta$) AND LIFETIME VALUE FUNCTION ($\phi$) UPDATE

To update the intrinsic reward parameters $\eta$, we directly take a meta-gradient ascent step using the overall objective (Equation 1). Specifically, the gradient is (see Appendix A for derivation):

$$\nabla_\eta J(\eta) = \mathbb{E}_{\theta_0\sim\Theta,\mathcal{T}\sim p(\mathcal{T})}\left[\mathbb{E}_{\tau_t\sim p(\tau_t|\eta,\theta_0)}\left[G^{\text{life}}_t\nabla_{\theta_t} \log \pi_{\theta_t}(a_t|s_t)\nabla_\eta\theta_t\right]\right], \qquad (3)$$

where $G^{\text{life}}_t = \sum_{k=t}^{T-1} \gamma^{k-t} r_{k+1}$ is a lifetime return based on the extrinsic rewards of task $\mathcal{T}$ with discount factor $\gamma$. The chain rule is used to get the meta-gradient ($\nabla_\eta\theta_t$) as in previous work (Zheng et al., 2018). The computation graph of this procedure is illustrated in Figure 1.

Computing the true meta-gradient in Equation 3 requires backpropagation through the entire lifetime, which is infeasible as each lifetime can involve more than thousands of policy updates. To partially address this issue, we truncate the meta-gradient after $N$ policy updates but approximate the lifetime return $G^{\text{life},\phi}_t \approx G^{\text{life}}_t$ using a *lifetime value function* $V_\phi(\tau)$ parameterised by $\phi$, which is learned using a temporal difference learning from $n$-step trajectory:

$$G^{\text{life},\phi}_t = \sum_{k=0}^{n-1} \gamma^k r_{t+k+1} + \gamma^n V_\phi(\tau_{t+n}) \qquad \phi = \phi + \alpha'(G^{\text{life},\phi}_t - V_\phi(\tau_t))\nabla_\phi V_\phi(\tau_t). \qquad (4)$$

In our empirical work, we found that the lifetime value estimates were crucial to allow the intrinsic reward to perform long-term credit assignments across episodes.

## 3.4 CONNECTION TO STANDARD RL FRAMEWORKS

The policy learning problem specified in Section 3.2 deviates from the standard Markov Decision Process (MDP) framework because the intrinsic reward function is a function of the lifetime history rather than a function of states or state-action pairs. Consequently, from the memoryless policy's perspective, the rewards are non-stationary. However, we can also view the combination of the intrinsic reward function and the policy as a joint lifetime-history-based policy parameterised by $\eta$ and $\theta$ (see derivation in Appendix A). From this perspective, the overall learning problem specified in Section 3.3 can be formulated as an MDP with history as state (recall, we use RNNs for the intrinsic reward function). As a result, standard temporal-difference learning methods are applicable to learning lifetime value functions.

## 4 EMPIRICAL INVESTIGATIONS: FEASIBILITY AND USEFULNESS

We present the results from our empirical investigations in two sections. For the results in this section, the experiments and domains are designed to answer the following research questions:

- What kind of knowledge can be learned by the intrinsic reward?
- How does the distribution of tasks drive the form of the intrinsic reward?
- What is the benefit of the lifetime return objective over the episode return?
- When is it important to provide the lifetime history as input to the intrinsic reward?

We investigate these research questions in various grid-world domains illustrated in Figure 2. For each domain, we trained an intrinsic reward function across many lifetimes and evaluated it by training an agent using the learned reward. We implemented the following baselines.

- Extrinsic-EP: A policy is trained with extrinsic rewards to maximise the episode return.

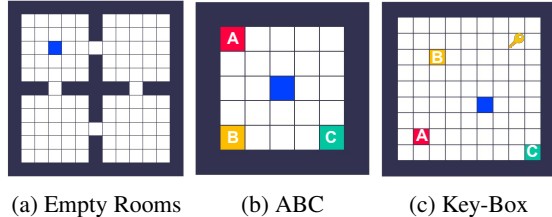

| (a) Empty Rooms | (b) ABC | (c) Key-Box |

Figure 2: Illustration of domains. (a) The agent needs to find the goal location which gives a positive reward, but the goal is not visible to the agent. (b) Each object (A, B, and C) gives rewards. (c) The agent is required to first collect the key and visit one of the boxes (A, B, and C) to receive the corresponding reward. All objects are placed to random locations after every episode.

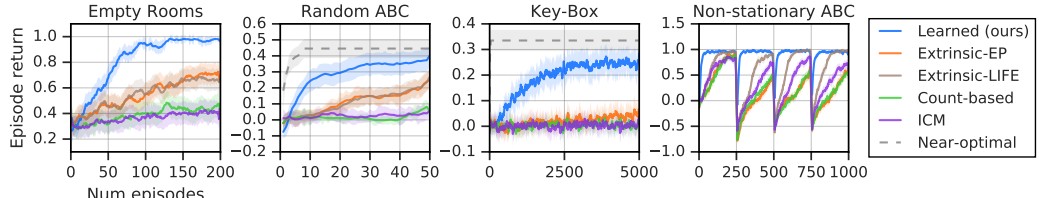

Figure 3: Evaluation of different reward functions averaged over 30 seeds. The learning curves show agents trained with our intrinsic reward (blue), with the extrinsic reward with the episodic return objective (orange) and the lifetime return objective (brown), and with a count-based exploration reward (green). The dashed line corresponds to a hand-designed near-optimal exploration strategy.

- Extrinsic-LIFE: A policy is trained with extrinsic rewards to maximise the lifetime return.

- Count-based (Strehl & Littman, 2008): A policy is trained with extrinsic rewards and count-based exploration bonus rewards.

- ICM (Pathak et al., 2017): A policy is trained with extrinsic rewards and curiosity rewards based on an inverse dynamics model.

Note that these baselines, unlike the learned intrinsic rewards, do not transfer any knowledge across different lifetimes. Throughout Sections 4.1-4.4, we focus on analysing what kind of knowledge is learned by the intrinsic reward depending on the nature of environments. We discuss the benefit of using the lifetime return and considering the lifetime history when learning the intrinsic reward in Section 4.5. The details of implementation and hyperparameters are described in Appendix B.

## 4.1 EXPLORING UNCERTAIN STATES

We designed 'Empty Rooms' (Figure 2a) to see whether the intrinsic reward can learn to encourage exploration of uncertain states like novelty-based exploration methods. The goal is to visit an invisible goal location, which is fixed within each lifetime but varies across lifetimes. Episode terminates when the goal is reached. Each lifetime consists of 200 episodes. From the agent's perspective, its policy should visit the locations suggested by the intrinsic reward. From the intrinsic reward's perspective, it should encourage the agent to go to unvisited locations to locate the goal, and once the goal is located to exploit that knowledge for the rest of that lifetime.

Figure 3 shows that our learned intrinsic reward was more efficient than extrinsic rewards and count-based exploration when training a new agent. We observed that the intrinsic reward learned two interesting strategies as visualised in Figure 4. While the goal is not found, it encourages exploration of unvisited locations, because it learned the prior that there exists a rewarding goal location somewhere. Once the goal is found the intrinsic reward encourages the agent to exploit it without further exploration, because it learned that there is only one goal. This result shows that curiosity about uncertain states can naturally emerge when various states can be rewarding in a domain, even when the rewarding states are fixed within an agent's lifetime.

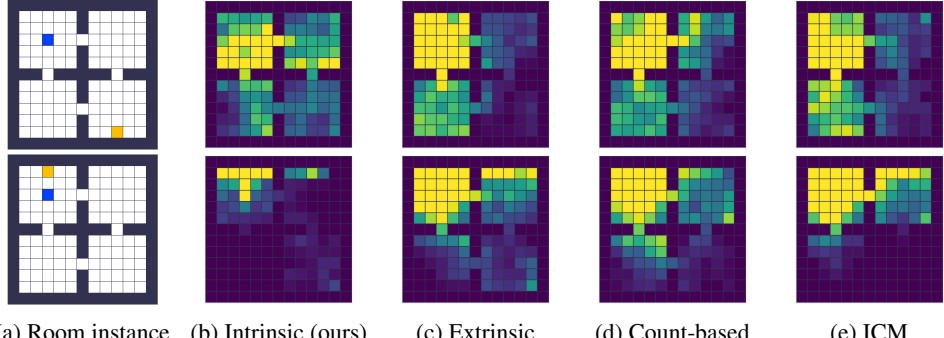

| (a) Room instance | (b) Intrinsic (ours) | (c) Extrinsic | (d) Count-based | (e) ICM |

Figure 4: Visualisation of the first 3000 steps of an agent trained with different reward functions in Empty Rooms. (a) The blue and yellow squares represent the agent and the *hidden* goal, respectively. (b) The learned reward encourages the agent to visit many locations if the goal is not found (top). However, when the goal is found early, the intrinsic reward makes the agent exploit it without further exploration (bottom). (c) An agent trained only with extrinsic rewards explores poorly. (d-e) Both the count-based and ICM rewards tend to encourage exploration (top) but hinders exploitation when the goal is found (bottom).

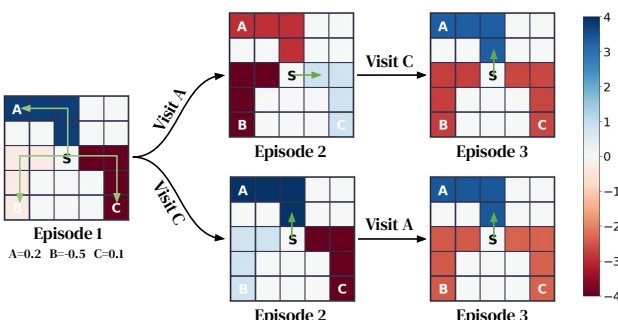

Figure 5: Visualisation of the learned intrinsic reward in Random ABC, where the extrinsic rewards for A, B, and C are 0.2, -0.5, and 0.1 respectively. Each figure shows the sum of intrinsic rewards for a trajectory towards each object (A, B, and C). In the first episode, the intrinsic reward encourages the agent to explore A. In the second episode, the intrinsic reward encourages exploring C if A is visited (top) or vice versa (bottom). In episode 3, after both A and C are explored, the intrinsic reward encourages to revisit A (both top and bottom).

## 4.2 EXPLORING UNCERTAIN OBJECTS AND AVOIDING HARMFUL OBJECTS

In the previous domain, we considered uncertainty of where the reward (or goal location) is. We now consider dealing with uncertainty about the value of different objects. In the 'Random ABC' environment (see Figure 2b), for each lifetime the rewards for objects A, B, and C are uniformly sampled from $[-1, 1]$, $[-0.5, 0]$, and $[0, 0.5]$ respectively but are held fixed within the lifetime. A good intrinsic reward should learn that: 1) B should be avoided, 2) A and C have uncertain rewards, hence require systematic exploration (first go to one and then the other), and 3) once it is determined which of the two A or C is better, exploit that knowledge by encouraging the agent to repeatedly go to that object for the rest of the lifetime.

Figure 3 shows that the agent learned a near-optimal exploration-and-then-exploitation method with the learned intrinsic reward. Note that the agent cannot pass information about the reward for objects across episodes, as usual in reinforcement learning. The intrinsic reward can propagate such information across episodes and help the agent explore or exploit appropriately. We visualised the learned intrinsic reward for different actions sequences in Figure 5. The intrinsic rewards encourage the agent to explore towards A and C in the first few episodes. Once A and C are explored, the agent exploits the largest rewarding object. Throughout training, the agent is discouraged to visit B through negative intrinsic rewards. These results show that avoidance and curiosity about uncertain objects can potentially emerge if the environment has various or fixed rewarding objects.

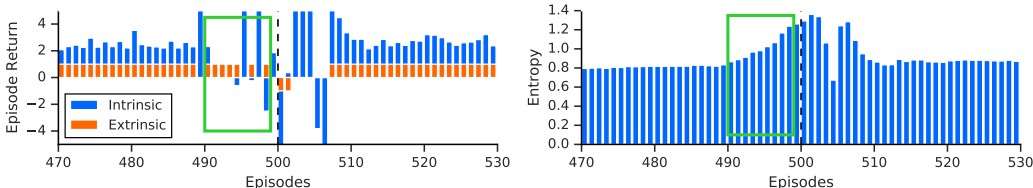

Figure 6: Visualisation of the agent's intrinsic and extrinsic rewards (left) and the entropy of its policy (right) on Non-stationary ABC. The task changes at 500th episode (dashed vertical line). The intrinsic reward gives a negative reward even before the task changes (green rectangle) and makes the policy less peaky (entropy increases). As a result, the agent quickly adapts to the change.

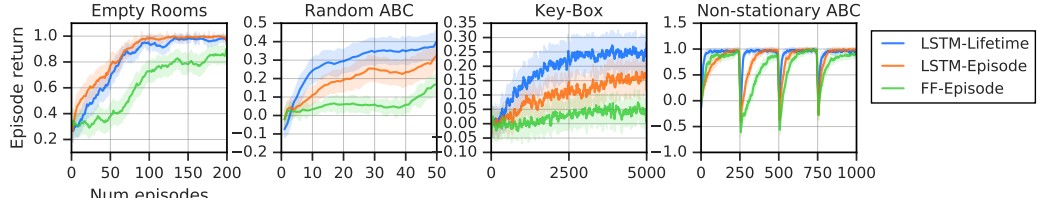

Figure 7: Evaluation of different intrinsic reward architectures and objectives. For 'LSTM' the reward network has an LSTM taking the lifetime history as input. For 'FF' a feed-forward reward network takes only the current time-step. 'Lifetime' and 'Episode' means the lifetime and episodic return as objective respectively.

## 4.3 Exploiting Invariant Causal Relationship

To see how the intrinsic reward deals with causal relationship between objects, we designed 'Key-Box', which is similar to Random ABC except that there is a key in the room (see Figure 2c). The agent needs to collect the key first to open one of the boxes (A, B, and C) and receive the corresponding reward. The rewards for the objects are sampled from the same distribution as Random ABC. The key itself gives a neutral reward of $0$. Moreover, the locations of the agent, the key, and the boxes are randomly sampled for each episode. As a result, the state space contains more than 3 billion distinct states and thus is infeasible to fully enumerate. Figure 3 shows that learned intrinsic reward leads to a near-optimal exploration. The agent trained with extrinsic rewards did not learn to open any box. The intrinsic reward captures that the key is necessary to open any box, which is true across many lifetimes of training. This demonstrates that the intrinsic reward can capture causal relationships between objects when the domain has this kind of invariant dynamics.

## 4.4 Dealing with Non-stationarity

We investigated how the intrinsic reward deals with non-stationarity of tasks within a lifetime in our 'Non-stationary ABC' environment. Rewards are as follows: for A is either $1$ or $-1$, for B is $-0.5$, for C is the negative value of the reward for A. The rewards of A and C are swapped every 250 episodes. Each lifetime lasts 1000 episodes. Figure 3 shows that the agent with the learned intrinsic reward quickly recovered its performance when the task changes, whereas the baselines take more time to recover. Figure 6 shows how the learned intrinsic reward encourages the learning agent to react to the changing rewards. Interestingly, the intrinsic reward has learned to prepare for the change by giving negative rewards to the exploitation policy of the agent a few episodes before the task changes. In other words, the intrinsic reward starts to discourage the agent to commit to the current best rewarding object, thereby increasing entropy in the current policy in anticipation of the change, eventually making it easier to adapt quickly. This shows that the intrinsic reward can capture the (regularly) repeated non-stationarity across many lifetimes and make the agent intrinsically motivated not to commit too firmly to a policy, in anticipation of changes in the environment.

## 4.5 Ablation Study

To study relative benefits of the proposed technical ideas, we conducted an ablation study 1) by replacing the long-term lifetime return objective ($G^{\text{life}}$) with the episodic return ($G^{\text{ep}}$) and 2) by restricting the input of the reward network to the current time-step instead of the entire lifetime history. Figure 7 shows that the lifetime history was crucial to achieve good performance. This is reasonable because all domains require some past information (e.g., current object rewards in Random ABC,

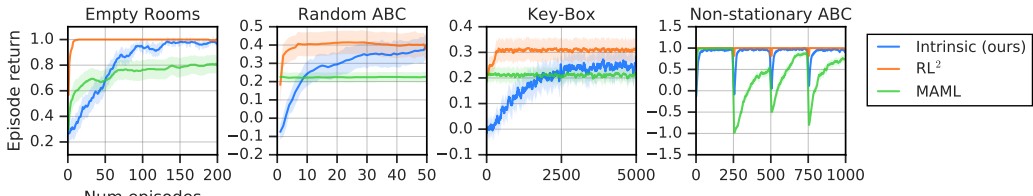

Figure 8: Comparison to policy transfer methods.

visited locations in Empty Rooms) to provide useful exploration strategies. It is also shown that the lifetime return objective was beneficial on Random ABC, Non-stationary ABC, and Key-Box. These domains require exploration across multiple episodes in order to find the optimal policy. For example, collecting an uncertain object (e.g., object A in Random ABC) is necessary even if the episode terminates with a negative reward. The episodic value function would directly penalise such an under-performed exploratory episode when computing meta-gradient, which prevents the intrinsic reward from learning to encourage exploration across episodes. On the other hand, such behaviour can be encouraged by the lifetime value function as long as it provides useful information to maximise the lifetime return in the long term.

## 5   EMPIRICAL INVESTIGATIONS: GENERALISATION VIA REWARDS

As noted above, rewards capture knowledge about what an agent's goals should be rather than how it should behave. At the same time, transferring the latter in the form of policies is also feasible in our domains presented above. Here we confirm that by implementing and presenting results for the following two meta-learning methods:

- MAML (Finn et al., 2017a): A policy meta-learned from a distributions of tasks such that it can adapt quickly to the given task after a few parameter updates.
- $RL^2$ (Duan et al., 2016; Wang et al., 2016): An LSTM policy unrolled over the entire lifetime to maximise the lifetime return, which is pre-trained on a distributions of tasks.

Although all the methods we implemented including ours are designed to learn useful knowledge from a distribution of tasks, they have different objectives. Specifically, the objective of our method is to learn knowledge that is useful for training "randomly-initialised policies" by capturing "what to do", whereas the goal of policy transfer methods is to directly transfer a useful policy for fast task adaptation by transferring "how to do" knowledge. In fact, it can be more efficient to transfer and reuse pre-trained policies instead of restarting from a random policy and learning using the learned rewards given a new task. Figure 8 indeed shows that $RL^2$ performs better than our intrinsic reward approach. It is also shown that MAML and $RL^2$ achieve good performance from the beginning, as they have already learned how to navigate the grid worlds and how to achieve the goals of the tasks. In our method, on the other hand, the agent starts from a random policy and relies on the learned intrinsic reward which indirectly tells it what to do. Nevertheless, our method outperforms MAML and achieves a comparable asymptotic performance to $RL^2$.

### 5.1   GENERALISATION TO DIFFERENT AGENT-ENVIRONMENT INTERFACES

In fact, our method can be interpreted as an instance of $RL^2$ with a particular decomposition of parameters ($\theta$ and $\eta$), which uses policy gradient as a recurrent update (see Figure 1). While this modular structure may not be more beneficial than $RL^2$ when evaluated with the same agent-environment interface, such a decomposition provides clear semantics of each module: the policy ($\theta$) captures "how to do" while the intrinsic reward ($\eta$) captures "what to do", and this enables interesting kinds of generalisations as we show below. Specifically, we show that "what" knowledge captured by the intrinsic reward can be reused by many different learning agents as follows.

**Generalisation to unseen action spaces**   We first evaluated the learned intrinsic reward on new action spaces. Specifically, the intrinsic reward was used to train new agents with either 1) permuted actions, where the semantics of left/right and up/down are reversed, or 2) extended actions, with 4 additional actions that move diagonally. Figure 9a shows that the intrinsic reward provided useful

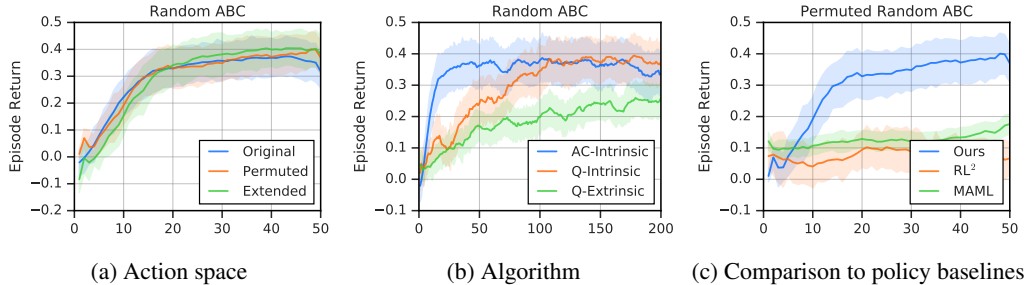

Figure 9: Generalisation to new agent-environment interfaces in Random ABC. (a) 'Permuted' agents have different action semantics. 'Extended' agents have additional actions. (b) 'AC-Intrinsic' is the original actor-critic agent trained with the intrinsic reward. 'Q-Intrinsic' is a Q-learning agent with the intrinsic reward learned from actor-critic agents. 'Q-Extrinsic' is the Q-learning agent with the extrinsic reward. (c) shows the performances of the policy transfer baselines with permuted actions during evaluation.

rewards to new agents with different actions, even when these were not trained with those actions. This is possible because the intrinsic reward assigns rewards to the agent's state changes rather than its actions. In other words, the intrinsic reward captures "what to do", which makes it possible to generalise to new actions, as long as the goal remains the same. On the other hand, it is unclear how to generalise $RL^2$ and MAML in this way.

**Generalisation to unseen learning algorithms**   We further investigated how general the knowledge captured by the intrinsic reward is by evaluating the learned intrinsic reward on agents with different learning algorithms. In particular, after training the intrinsic reward from actor-critic agents, we evaluated it by training new agents through Q-learning while using the learned intrinsic reward as denoted by 'Q-Intrinsic' in Figure 9b. Interestingly, it turns out that the learned intrinsic reward is general enough to be useful for Q-learning agents, even though it was trained for actor-critic agents. Again, it is unclear how to generalise $RL^2$ and MAML in this way.

**Comparison to policy transfer**   While it wasn't possible to apply the learned policy from $RL^2$ and MAML when we extended the action space and when we changed the learning algorithm, we can do so when we keep the same number of actions and just permute them. As shown in Figure 9c, both $RL^2$ and MAML generalise poorly when the action space is permuted for Random ABC, because the transferred policies are highly biased to the original action space. Again, this result highlights the difference between "what to do" knowledge captured by our approach and "how to do" knowledge captured by policies.

## 6   CONCLUSION

We revisited the optimal reward problem (Singh et al., 2009) and proposed a more scalable gradient-based method for learning intrinsic rewards. Through several proof-of-concept experiments, we showed that the learned non-stationary intrinsic reward can capture regularities within a distribution of environments or, over time, within a non-stationary environment. As a result, they were capable of encouraging both exploratory and exploitative behaviour across multiple episodes. In addition, some task-independent notions of intrinsic motivation such as curiosity emerged when they were effective for the distribution over tasks across lifetimes the agent was trained on. We also showed that the learned intrinsic rewards can generalise to different agent-environment interfaces such as different action spaces and different learning algorithms, whereas policy transfer methods fail to generalise. This highlights the difference between the "what" kind of knowledge captured by rewards and the "how" kind of knowledge captured by policies. The flexibility and range of knowledge captured by intrinsic rewards in our proof-of-concept experiments encourages further work towards combining different loci of knowledge to achieve greater practical benefits.

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

## A    DERIVATION OF INTRINSIC REWARD UPDATE

Following the conventional notation in RL, we define $v_\mathcal{T}(\tau_t|\eta,\theta_0)$ as the state-value function that estimates the expected future lifetime return given the lifetime history $\tau_t$, the task $\mathcal{T}$, initial policy parameters $\theta_0$ and the intrinsic reward parameters $\eta$. Specially, $v_\mathcal{T}(\tau_0|\eta,\theta_0)$ denotes the expected lifetime return at the starting state, i.e.,

$$v_\mathcal{T}(\tau_0|\eta,\theta_0) = \mathbb{E}_{\tau \sim p_\eta(\tau|\theta_0)}\left[G^{\text{life}}\right],$$

where $G^{\text{life}}$ denotes the lifetime return in task $\mathcal{T}$. We also define the action-value function $q_\mathcal{T}(\tau_t, a_t|\eta, \theta_0)$ accordingly as the expected future lifetime return given the lifetime history $\tau_t$ and an action $a_t$.

The objective function of the optimal reward problem is defined as:

$$J(\eta) = \mathbb{E}_{\theta_0 \sim \Theta, \mathcal{T} \sim p(\mathcal{T})}\left[\mathbb{E}_{\tau \sim p_\eta(\tau|\theta_0)}\left[G^{\text{life}}\right]\right] \tag{5}$$
$$= \mathbb{E}_{\theta_0 \sim \Theta, \mathcal{T} \sim p(\mathcal{T})}\left[v_\mathcal{T}(\tau_0|\eta, \theta_0)\right], \tag{6}$$

where $\Theta$ and $p(\mathcal{T})$ are an initial policy distribution and a task distribution respectively.

Assuming the task $\mathcal{T}$ and the initial policy parameters $\theta_0$ are given, we omit $\mathcal{T}$ and $\theta_0$ for the rest of equations for simplicity. Let $\pi_\eta(\cdot|\tau_t) = \pi_{\theta_t}(\cdot|s_t)$ be the probability distribution over actions at time $t$ given the history $\tau_t$, where $\theta_t = f_\eta(\tau_t, \theta_0)$ is the policy parameters at time $t$ in the lifetime. We can derive the meta-gradient with respect to $\eta$ by the following:

$$\nabla_\eta J(\eta)$$
$$= \nabla_\eta v(\tau_0|\eta)$$
$$= \nabla_\eta \left[\sum_{a_0} \pi_{\theta_0}(a_0|\tau_0) q(\tau_0, a_0|\eta)\right]$$
$$= \sum_{a_0} \left[\nabla_\eta \pi_{\theta_0}(a_0|\tau_0) q(\tau_0, a_0|\eta) + \pi_{\theta_0}(a_0|\tau_0)\nabla_\eta q(\tau_0, a_0|\eta)\right]$$
$$= \sum_{a_0} \left[\nabla_\eta \pi_{\theta_0}(a_0|\tau_0) q(\tau_0, a_0|\eta) + \pi_{\theta_0}(a_0|\tau_0)\nabla_\eta \sum_{\tau_1, r_0} p(\tau_1, r_0|\tau_0, a_0)\Big(r_0 + v(\tau_1|\eta)\Big)\right]$$
$$= \sum_{a_0} \left[\nabla_\eta \pi_{\theta_0}(a_0|\tau_0) q(\tau_0, a_0|\eta) + \pi_{\theta_0}(a_0|\tau_0)\sum_{\tau_1} p(\tau_1|\tau_0, a_0)\nabla_\eta v(\tau_1|\eta)\right]$$
$$= \mathbb{E}_{\tau_t}\left[\sum_{a_t} \nabla_\eta \pi_\eta(a_t|\tau_t) q(\tau_t, a_t|\eta)\right]$$
$$= \mathbb{E}_{\tau_t}\left[\nabla_\eta \log \pi_\eta(a_t|\tau_t) q(\tau_t, a_t|\eta)\right]$$
$$= \mathbb{E}_{\tau_t}\left[G_t \nabla_\eta \log \pi_\eta(a_t|\tau_t)\right]$$
$$= \mathbb{E}_{\tau_t}\left[G_t \nabla_{\theta_t} \log \pi_{\theta_t}(a_t|s_t)\nabla_\eta \theta_t\right],$$

where $G_t = \sum_{k=t}^{T-1} r_k$ is the lifetime return given the history $\tau_t$, and we assume the discount factor $\gamma = 1$ for brevity. Thus, the derivative of the overall objective is:

$$\nabla_\eta J(\eta) = \mathbb{E}_{\theta_0 \sim \Theta, \mathcal{T} \sim p(\mathcal{T})}\left[\mathbb{E}_{\tau_t \sim p(\tau_t|\eta, \theta_0)}\left[G_t \nabla_{\theta_t} \log \pi_{\theta_t}(a_t|s_t)\nabla_\eta \theta_t\right]\right]. \tag{7}$$

# B   EXPERIMENTAL DETAILS

## B.1   IMPLEMENTATION DETAILS

We used mini-batch update to reduce the variance of meta-gradient estimation. Specifically, we ran $64$ lifetimes in parallel, each with a randomly sample task and randomly initialised policy parameters. We took the average of the meta-gradients from each lifetime to compute the update to the intrinsic reward parameters($\eta$). We ran $2 \times 10^5$ updates to $\eta$ at training time. We used arctan activation on the output of the intrinsic reward. The hyperparameters used for each domain are described in Table 1.

Table 1: Hyperparameters.

| Hyperparameters | Empty Rooms | Random ABC | Key-Box | Non-stationary ABC |
|---|---|---|---|---|
| Time limit per episode | 100 | 10 | 100 | 10 |
| Number of episodes per lifetime | 200 | 50 | 5000 | 1000 |
| Inner unroll length | 8 | 4 | 16 | 4 |
| Entropy regularisation | 0.01 | 0.01 | 0.01 | 0.05 |
| Policy architecture | | Conv(16)-FC(64) | | |
| Policy optimiser | SGD | SGD | Adam | SGD |
| Policy learning rate | 0.1 | 0.1 | 0.001 | 0.1 |
| Reward architecture | | Conv(16)-FC(64)-LSTM(64) | | |
| Reward optimiser | | Adam | | |
| Reward learning rate | | 0.001 | | |
| Outer unroll length | | 5 | | |
| Inner discount factor | | 0.9 | | |
| Outer discounter factor | | 0.99 | | |

## B.2   DOMAINS

We will consider five task distributions, instantiated within one of the three main gridworld domains shown in Figure 2. In all cases the agent has four actions available, corresponding to moving up, down, left and right. However the topology of the gridworld and the reward structure may vary.

### B.2.1   EMPTY ROOMS

Figure 2a shows the layout of the *Empty Rooms* domain. There are four rooms in this domain. The agent always starts at the centre of the top-left room. One and only one cell is rewarding, which is called the goal. The goal is invisible. The goal location is sampled uniformly from all cells at the beginning of each lifetime. An episode terminates when the agent reaches the goal location or a time limit of 100 steps is reached. Each lifetime consists of 200 episodes. The agent needs to explore all rooms to find the goal and then goes to the goal afterwards.

### B.2.2   ABC WORLD

Figure 2b shows the layout of the *ABC World* domain. There is a single 5 by 5 room, with three objects (denoted by A, B, C). All object provides reward upon reaching them. An episode terminates when the agent reaches an object or a time limit of 10 steps is reached. We consider three different versions of this environment: *Fixed ABC*, *Random ABC* and *Non-stationary ABC*. In the Fixed ABC environment, each lifetime has 200 episodes. The reward associated with each object is fixed across lifetimes. Specifically, the rewards for objects A, B, and C are $1$, $-0.5$, and $0.5$ respectively. The optimal policy is to always collect A. In the Random ABC environment, each lifetime has 50 episodes. The reward associated with each object is randomly sampled for each lifetime and is held fixed within a lifetime. Thus, the environment is stationary from an agent's perspective but non-stationary from the reward function's perspective. Specifically, the rewards for A, B, and C are uniformly sampled from $[-1, 1]$, $[-0.5, 0]$, and $[0.0.5]$ respectively. The optimal behaviour is to explore A and C at the beginning of a lifetime to assess which is the better, and then commits to the better one for all subsequent episode. In the non-stationary ABC environment, each lifetime has

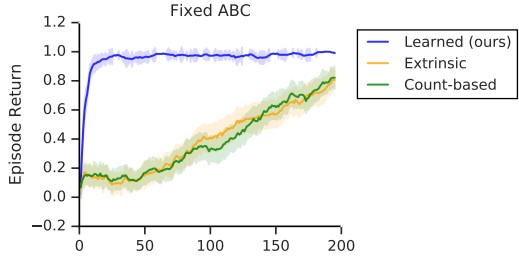

Figure 10: Evaluation of different rewards in the Fixed ABC domain. The x-axis shows the number of episodes within a single lifetime; the y-axis measures the episode return.

1000 episodes. The rewards for A, B, and C are $1$, $-0.5$, and $-1$ respectively. The rewards for A and C swap every 250 episodes.

### B.2.3    KEY BOX WORLD

Figure 2c shows the *Key Box World* domain. In this domain, there is a key and three boxes, A, B, and C. In order to open any box, the agent must pick up the key first. The rewards for A, B, and C are uniformly sampled from $[-1, 1]$, $[-0.5, 0]$, and $[0, 0.5]$ respectively for each lifetime. An episode terminates when the agent opens a box or a time limit of $100$ steps is reached. Each lifetime consists of $5000$ episodes.

### B.3    HAND-DESIGNED NEAR-OPTIMAL EXPLORATION STRATEGY FOR RANDOM ABC

We hand-designed a heuristic strategy for the Random ABC domain. We assume the agent has the prior knowledge that B is always bad and A and C have uncertain rewards. Therefore, the heuristic is to go to A in the first episode, go to C in the second episode, and then go to the better one in the remaining episodes in the lifetime. We view this heuristic as an upper-bound because it always finds the best object and can arbitrarily control the agent's behaviour.

## C    ADDITIONAL EMPIRICAL RESULT

### C.1    EXPLOITING OPTIMAL BEHAVIOUR ON A FIXED TASK

To investigate what the intrinsic reward learns in a fixed task, we designed the 'Fixed ABC' environment (see Figure 2b). The reward for each object (A, B, and C) is fixed within and across lifetimes. When the agent collects an object, it receives the corresponding reward of $1$, $-0.5$, or $0.5$ for object A, B, or C respectively, and the episode terminates. Each lifetime contains $200$ episodes. The optimal policy is to always collect A. The optimal reward should capture the regularity of the environment that object A has the highest reward and drive the agent towards object A.

Figure 10 shows that agents trained with the learned intrinsic reward learn optimal policies within a few episodes. This indicates that the intrinsic reward memorises the fixed optimal behaviour during training and assigns rewards accordingly to aid learning during evaluation. The result on Fixed ABC is not particularly surprising. In a fixed task in a stationary environment, an optimal reward function does not need to encourage exploration, and helps the agent to directly learn the optimal behaviour as quickly as possible, similarly to reward shaping.

