# OpenReview forum: "What Can Learned Intrinsic Rewards Capture?"
_ICLR.cc/2020/Conference — Reject_

### Official Review · AnonReviewer2 · 2019-10-20
**Official Blind Review #2**

**Rating:** 6

**Review:**

Summary
The paper evaluates the intrinsic reward as a way of storing information about episodes. It adopts the optimal intrinsic reward setting (Singh'09), and extends its recent policy gradient implementation, LIRPG, to lifetime settings. The task in the lifetime setting is to learn an intrinsic reward such that when trained with it, the agent maximizes its total return over its lifetime. A lifetime is defined as a sequence of episodes, where the agent does not have memory of previous episodes, however, the function computing the intrinsic reward does. In proof-of-concept experiments, the paper demonstrates that the learned intrinsic reward captures properties of several gridworld environments and induces meaningful behavior in the agent, successfully transferring information from previous episodes. Interestingly, a state-based reward function also generalizes to agents with perturbed action spaces, showing that this way of storing information is agnostic to the agent’s action space.

Decision
The paper proposal is interesting and adequately evaluated, however, the impact of the paper might be limited by its limited technical novelty and lack of comparisons to strong baselines. I recommend marginal accept.

Pros
- The paper is well-motivated.
- The paper is well-written and the method is clearly explained. The literature review is thorough.
- The experimental evaluation demonstrates several interesting and potentially promising phenomena.

Cons
- The novelty of the paper is limited as it is a somewhat straightforward extension of prior work.
- The impact of the paper is hard to judge as the experimental evaluation does not focus on potential usecases.

Questions. Here, I will focus on scientific questions, answering which would significantly improve the quality of the paper.
- The biggest drawback of the paper is that the proposed method has an unfair advantage as it has a way of transmitting information across episodes, which the baselines do not (as stated on the bottom of page 5). While the findings of this paper are interesting, it is unclear how it compares to methods that have memory of previous episodes, such as agents with non-episodic recurrent policies, or meta-learning agents such as Duan’16, Finn’17. Is it possible that the proposed method e.g. scales better than recurrent policies due to compact representations or provides better generalization to things like action space changes?
- How does the method compare to hand-designed intrinsic rewards on hard exploration games (such as montezuma’s revenge or pitfall Atari games)? Since it can only learn to explore on games that it previously successfully solved, it is possible that a hand-designed intrinsic reward such as RND (Burda’19) would perform better on these hard games. On the other hand, it is possible that the method will in fact perform better on these games due to more directed exploration.
- How does the method compare to hand-designed intrinsic reward on out-of-distribution tasks? Intuitively, the method should perform the worse the further from the training distribution the task is, while the hand-designed rewards will always perform similarly. However, what is the extent to which the proposed method generalizes? It is possible that this method would be very useful in practice if it generalized well.

Other potentially related work.
- Xu’18, Learning to Explore with Meta-Policy Gradient, is a relevant work that proposes a meta-learning framework for training an exploration policy.
- Metz’19, Meta-Learning Update Rules for Unsupervised Representation Learning, is a conceptually relevant work that proposes to meta-learn loss functions for unsupervised learning (and there is more recent related work on this topic too).


**Experience Assessment:**

I have read many papers in this area.

**Review Assessment: Checking Correctness Of Derivations And Theory:**

I assessed the sensibility of the derivations and theory.

**Review Assessment: Checking Correctness Of Experiments:**

I carefully checked the experiments.

**Review Assessment: Thoroughness In Paper Reading:**

I read the paper thoroughly.

---

> ### Author Response · Authors · 2019-11-12
> **Response to R2**
>
> Thank you very much for constructive comments. We address the questions below and reflected some of the suggestions in the revision (see the common response above).
>
> # Regarding comparison to other meta-learning methods
> We added a comparison to two meta-learning methods (RL^2 and MAML). Please see the details in the common response (see Section 5).
>
> # Regarding comparison to hand-designed intrinsic rewards on hard exploration problems
> The goal of this paper is to show that interesting kinds of “what” knowledge can be captured by learned intrinsic rewards such as exploring uncertainty and provide in-depth analysis of the approach. We would like to explore scaling to hard exploration tasks like Montezuma’s Revenge as future work.
>
> # Regarding comparison to hand-designed intrinsic rewards on out-of-distribution tasks
> We demonstrated that the intrinsic reward can interpolate successfully within the same task distribution. However, it is unclear whether it can extrapolate to out-of-distribution tasks, as the neural network representation should successfully handle extrapolation, which is an active research topic in deep learning (e.g., disentangled representation). We believe that more research including representation learning is needed to learn intrinsic rewards that can generalise well to out-of-distribution tasks. We would like to investigate in this direction in the future.
>
> # Regarding missing references
> We added missing references mentioned by the reviewer in the revision.

---

> > ### Comment · AnonReviewer2 · 2019-11-12
> > **Thanks for the thorough response!**
> >
> > The authors addressed my original concerns and performed a significant amount of additional work to strengthen the experimental results. In particular, I appreciate the new findings that 1) the method is able to generalize to a different RL algorithm, which seems close to impossible with MAML-like approaches, 2) the method outperforms MAML and RL^2 on generalization to new action spaces, and 3) it also outperforms MAML overall, which could perhaps be stressed more in the paper :). I believe that the paper is much improved after the update and I am now entirely convinced the paper should be accepted.
> >
> > --------------------------------------------------------------------
> > Additional comments
> >
> > In general, I am under the impression that in relation to RL^2 this paper represents a more constrained approach, which adapts slower as it is less expressive, but should also generalize better. This opens the following next questions (and perhaps more of them), answering which would strengthen the paper even more.
> > - How does the method compare to hand-designed intrinsic rewards like Ostrovsky'17, Pathak'17 on the current tasks (as well as on the tasks suggested in my original review)?
> > - Does the method improve over approaches designed for changing action spaces like Devin'16 or Chandak'18?
> > - More generally, if the rewards can be a locus of knowledge, how can this knowledge be used? One could perhaps think of some kind of distillation or guided policy search mechanism, although this is just a raw thought I have.
> >
> > Chandak et al., Reinforcement Learning with a Dynamic Action Set, 2018
> > Devin et al., Learning Modular Neural Network Policies for Multi-Task and-Robot Transfer, 2016

---

### Official Review · AnonReviewer1 · 2019-10-20
**Official Blind Review #1**

**Rating:** 6

**Review:**


The paper proposes a meta-learning approach to learn reward functions for reinforcement learning agents. It defines an algorithm to optimize an intrinsic reward function for a distribution of tasks in order to maximise the agent’s lifetime rewards. The properties of this reward function and meta-learning algorithm  are investigated through a number of proof-of-concept experiments.

The meta-learning algorithm and the corresponding empirical investigation are the main contributions of the paper. The algorithm seems to be similar to previous meta-learning approaches, but differs by introducing a lifetime value function. While I thought the paper raises some interesting possibilities, I am currently leaning towards rejection. The proposed algorithm does not seem like a major innovation over cited previous work. The empirical evaluation provides a number of proof-of-concept ideas, but no in depth investigation of the properties of the approach. The theoretical properties of the approach are barely discussed.

Detailed remarks:

* The main addition to the meta-learning algorithm is the lifetime value function. The authors mention multiple times that this is crucial to learning, but the properties of this value function are not really investigated or discussed in depth:

- The authors mention that the value function must take into account changing future policies, but do not discuss this further. The value function update seems to be a standard on-policy TD update with the lifetime return and the complete history as input. The policy for this value function, however, is still a standard policy with only state as input (but it will be non-stationary over the agent lifetime). It would be good to discuss this learning problem in more detail.
- The algorithm uses an n-step return. Is this important? What effect does n have on learning?

* Another issue which I would have liked being discussed in more detail is the non-stationarity of the learning problem in general. Most of the approaches discussed in related work (e.g. shaping)  are aimed at learning/designing more informative reward functions. These reward functions still fit in the MDP framework, however, and map from states and actions to rewards. In the case of shaping approaches guarantees can be given that this does not alter the learning problem. The intrinsic reward functions used in this paper map the full life-time history of the agent to rewards. While this is a richer framework that can express more complicated tasks (like exploration over multiple episodes), it also invalidates many of the basic assumptions of reinforcement learning. The rewards are now no longer Markovian when only observing the current state. Moreover, the reward function will change over time. To what extent does this require non-stationary / history-based policy and value function learning to solve these issues? While some of these issues also apply to count based exploration strategies, (Strehl and Littman,2008 ) provided results that the  exploration bonuses result a Bellman Equation that accounts for uncertainties. No real guarantees seem to exist here.

* The empirical contribution focuses on trying to answer a number of questions regarding the properties of the learnt intrinsic rewards. I found these questions to be very broad, while the answers are mostly anecdotal evidence through proof-of-concept examples.  These examples do show potential benefits of meta-learning intrinsic rewards, but I was somewhat disappointed that there was no more systematic investigation. For example, questions like ‘how does the distribution of tasks affect intrinsic rewards’ or ‘does intrinsic reward generalise’ are not really answered by providing metrics of performance or generalisation in controlled experiments, but by providing some example cases. Several of these questions (including optimising exploration and dealing with non-stationarity) also seem to have been investigated to some extent in the original Optimal reward papers (Singh, 2009/2010). It would be good to clearly indicate what we have learned beyond these previous results.

* There seems to be a bit of a mismatch between the learning objective for intrinsic rewards in the optimal reward framework and the results shown in the experiments. The learning objective aims to optimise lifetime rewards for a distribution of tasks. Most of the experiments seem to analyse episodic reward performance and compare against single-task (or task agnostic) methods.

Minor comments:

- The architecture / parameterization of the lifetime value function does not seem to be defined anywhere. Given that it takes histories as input I assume this is another RNN?
- There seems to be some small overloading in the notation with \eta occasionally being used to denote the parameters of the reward function r_eta or the reward function itself.


**Experience Assessment:**

I have published in this field for several years.

**Review Assessment: Checking Correctness Of Derivations And Theory:**

I assessed the sensibility of the derivations and theory.

**Review Assessment: Checking Correctness Of Experiments:**

I carefully checked the experiments.

**Review Assessment: Thoroughness In Paper Reading:**

I read the paper thoroughly.

---

> ### Author Response · Authors · 2019-11-12
> **Response to R1**
>
> Thank you very much for constructive comments. We address the questions below and reflected some of the suggestions in the revision (see the common response above).
>
> # Regarding the non-stationary learning problem and theoretical guarantee
> As the reviewer pointed out, the problem is indeed non-stationary from the memoryless policy’s perspective. However, we can also view the combination of the intrinsic reward function and the policy as a joint lifetime-history-based policy parameterised by $\eta$ and $\theta$ (see derivation in Appendix A). From this perspective, the overall learning problem can be formulated as an MDP with history as state (recall, we use RNNs for the intrinsic reward function). We revised the paper to make this  point clear. (see Section 3.4)
>
> # Regarding systematic investigation of the learned intrinsic rewards
> We showed that the intrinsic reward captures quite different but appropriate knowledge by varying reward functions in ABC domain (i.e., Fixed ABC in Figure 10 → Random ABC → Non-stationary ABC). We agree that further systematic investigation could help and would appreciate if the reviewer makes a concrete suggestion on this.
>
> # Regarding what we learned beyond previous work
> We revised the abstract to further highlight our contribution. Specifically, we learned the following beyond previous work as follows. (1) It is possible to learn good reward functions via gradient-based meta-learning, which is much more scalable than exhaustive search (prior work). (2) The meta-learned reward functions can capture interesting kinds of ``what'' knowledge, which includes long-term exploration and exploitation. (3) Because of the indirectness of this form of knowledge the learned reward functions can generalise to other kinds of agents and to changes in the dynamics of the environment.
>
> # Regarding mismatch between the learning objective and the experimental results
> The objective for training the intrinsic reward function is to maximise cumulative lifetime rewards. By looking at the area-under-the-curve in our evaluation results, we can observe lifetime rewards. Thus, we believe that the evaluation curves show both metrics (i.e., episodic return and lifetime return). Our paper also acknowledges that the baseline reward functions are task-independent (Section 4).
>
> # Regarding missing architecture details and overloaded notations
> We added some missing details about the lifetime value function architecture and revised the notations in the revised paper.

---

### Official Review · AnonReviewer3 · 2019-10-24
**Official Blind Review #3**

**Rating:** 6

**Review:**

(Originally my score was a weak reject.)

This paper aims to study whether a learned reward function can serve as a locus of knowledge about the environment, that can be used to accelerate training of new agents. The authors create an algorithm that learns an intrinsic reward function, that when used to train a new agent over a “lifetime” (which consists of multiple episodes), leads to the best cumulative reward over the lifetime. As a result, the learned intrinsic reward is incentivized to quickly “teach” the agent when and where to explore to find out as-yet unknown information, and then exploit that information once there is no more to be had. Experiments on gridworlds demonstrate that these learned intrinsic rewards: 1. switch between early exploration and later exploitation, 2. explore only for information that is relevant for optimal behavior, 3. capture invariant causal relationships, and 4. can anticipate and adapt to changes in the extrinsic reward within a lifetime.

I very much appreciated the design of the environments to test for specific properties within the learning algorithm: I think these experiments provide a very useful conceptual analysis of what learned intrinsic rewards can do.

My main qualm with the paper is with its significance -- the authors claim that the goal is to find out whether reward functions can be loci of knowledge, but we already know the answer is yes: the whole point of reward shaping is to improve training dynamics by building in knowledge into the reward function. It is not a surprise that learned reward functions can be loci of knowledge if our hand-designed reward functions already do so.

To me, the more interesting aspect of this paper is how much benefit we can get by learning intrinsic reward functions, relative to other ways of improving training dynamics. The authors do show that by allowing the intrinsic reward to be recurrent (and so dependent on past episodes), it is able to first incentivize exploration and later exploitation, which standard reward shaping cannot do (since usually reward shaping still maintains the assumption that the reward is a function of the state). However, given this motivation, it would be important to see comparisons between the proposed method of learning intrinsic rewards, and other methods for fast adaptation in the literature, such as MAML, which as I understand also has many of the properties highlighted in this paper.

Ideally there would also be experiments on more complex environments: the environments in the paper have 104, 25, and 49 states. If we in the ABC environments if you count “whether or not reward(object) is known” as part of the state, that multiplies it by 2^3 = 8 giving 200 and 392 states, if you then further add the ordering of r(A), r(B), and r(C), that multiplies by a factor of 3! = 6 giving 1200 and 2352 states. These environments are excellent for demonstrating the properties of learned intrinsic rewards and I am glad the authors have done these experiments and analyzed the results. However, given that the paper aims to scale the optimal reward problem, it would have been useful to see examples where the state space cannot be fully enumerated to evaluate scalability.

Questions:

In Figure 5, in episode 1, why is the learned intrinsic reward heavily penalizing the path to C, but not penalizing the path to B? In the initial episode, the intrinsic reward should only know that B is to be avoided; it doesn’t yet know whether A or C is the better object.  I would expect the learned intrinsic reward to put similar positive rewards on the path to C and the path to A, and negative reward on the path to B. (It is slightly more likely that C is the best object. This probably changes things slightly, but not significantly.)

Also in Figure 5, by episode 3, shouldn’t the final states (A or C) have intrinsic rewards of larger magnitude? Otherwise the agent can go back and forth on the path to collect lots of intrinsic reward without terminating the episode, even though this wouldn’t get extrinsic reward.

**Experience Assessment:**

I have published one or two papers in this area.

**Review Assessment: Checking Correctness Of Derivations And Theory:**

I assessed the sensibility of the derivations and theory.

**Review Assessment: Checking Correctness Of Experiments:**

I assessed the sensibility of the experiments.

**Review Assessment: Thoroughness In Paper Reading:**

I read the paper at least twice and used my best judgement in assessing the paper.

---

> ### Author Response · Authors · 2019-11-12
> **Response to R3**
>
> Thank you very much for constructive comments. We address the questions below and reflected some of the suggestions in the revision (see the common response above).
>
> # Regarding “the goal is to find out whether reward functions can be loci of knowledge”
> We clarify that our goal is not just finding out whether it is possible to store knowledge into rewards. In fact, we acknowledge in the introduction that existing hand-designed rewards already show that they can be a locus of knowledge. Instead, our goal is to find out 1) whether it is feasible to capture knowledge in reward functions in a data-driven from the agent’s own experience rather than hand-designing them, 2) what kind of knowledge can be captured when they are “learned” rather than “hand-designed”, and 3) to show that reward knowledge can generalise to new dynamics and new learning algorithms. We clarified this in the revision.
>
> # Regarding the benefits of learning intrinsic rewards in comparison to other methods
> In Section 5, we added a comparison to RL^2 and MAML and added one more experiment demonstrating that the intrinsic rewards learned from actor-critic agents can generalise to a different kind of learning agents, i.e. Q-learning agents. Please see the common response for details.
>
> # Regarding more complex domains
> We revised the paper with a new version of the key-box domain, where the map is a 9x9 grid world and objects are randomly placed for each episode. Due to the random placement, there are more than 3 billion distinct states. We acknowledge that this number is still tiny in comparison to domains with high-dimensional visual observations, but this shows that our method can scale up to larger domains, where it is infeasible to fully enumerate the entire state space.
>
> # Regarding questions about Figure 5
> Regarding your question about episode 1, we conjecture that it is more optimal for the intrinsic reward to encourage the agent to commit to one particular object (either A or C) at the beginning of training. Otherwise, if the reward is equal for A and C, it would take more time for a “randomly-initialised” policy to learn to collect any of them, because going towards both objects are encouraged (and they are placed in the opposite positions).
> Regarding your question about episode 3, the colors represent the return for each trajectory not per-step reward. Therefore, the agent would not gain more rewards by moving back and forth. Also, it is important to note that the intrinsic reward is a function of the agent’s history. So, it is very likely that the intrinsic reward would penalise if the agent keeps going back and forth without proper exploration/exploitation, which would be an interesting analysis to be done.

---

> > ### Comment · AnonReviewer3 · 2019-11-13
> > **Thanks!**
> >
> > Thanks for the response!
> >
> > I especially liked the increased focus on the intrinsic reward as allowing us to separate what to do from how to do it, and showing that this allows for generalization across action spaces, learning algorithms, etc. This is a good point that I hadn't thought about before.
> >
> > With the new experiment with the key-box experiment, what reward would the optimal policy get? I would assume that it should be the same as in Random ABC, but the learned policies do significantly worse, which might mean that there are problems scaling up the method. (Nonetheless, it still seems to get close to RL^2 and is better than MAML, so probably this is normal.)
> >
> > > Regarding your question about episode 3, the colors represent the return for each trajectory not per-step reward. Therefore, the agent would not gain more rewards by moving back and forth.
> >
> > Oh, that makes much more sense.
> >
> > > Regarding your question about episode 1, we conjecture that it is more optimal for the intrinsic reward to encourage the agent to commit to one particular object (either A or C) at the beginning of training. Otherwise, if the reward is equal for A and C, it would take more time for a “randomly-initialised” policy to learn to collect any of them, because going towards both objects are encouraged (and they are placed in the opposite positions).
> >
> > This also makes more sense now that I realize the colors were just about the trajectories -- probably the rewards are positive if going towards A and negative when going away from A, which incidentally makes B be ~zero reward while C is negative.
> >
> > Mostly due to the explanation of the significance of learned intrinsic rewards, I'm changing to a weak accept. (I still have some qualms about scalability, though the new key-box environments has addressed that somewhat.)

---

### Author Response · Authors · 2019-11-12
**Common Response**

Thank you very much for constructive comments. We address some common questions below.

# Paper revision
We significantly revised the paper based on the reviewers’ suggestions as follows.
- We added a comparison to RL^2 and MAML in Section 5 and Figure 8 and an in-depth discussion about the difference between our approach and policy transfer approaches.
- To further highlight the difference between our approach and policy transfer methods, we added one more experiment demonstrating that the intrinsic rewards learned from actor-critic agents can generalise to different types of learning agents, i.e. Q-learning agents.
- To address the concern regarding the small state spaces, we replaced the Key-Box domain with a more challenging version, where objects are randomly placed for each episode in a larger map (5x5 to 9x9). This results in a significantly larger state space due to its combinatorial nature. (see Figure 2c and Figure 3)
- We addressed several writing issues including missing references, missing architecture details, overloaded notations, etc.

# Regarding comparison to MAML and RL^2
We added a comparison to RL^2 and MAML (see Section 5 and Figure 8 and Figure 9). We emphasise that the goal of our method and the goals of RL^2 and MAML are different. To summarise, our method performs better than MAML but learns slowly compared to RL^2 (with similar asymptotic performance). However, we also show that these policy transfer methods generalise poorly to unseen action-environment interfaces (or not capable of generalisation in most cases), whereas the intrinsic reward can successfully generalise. This highlights the difference between “what to do” knowledge captured by intrinsic rewards and “how to do” knowledge captured by policy transfer methods. We would appreciate if the reviewers take a look at the revised paper for more in-depth discussion (Section 5).

# Regarding significance
Although some techniques used in our paper have connections to existing meta-learning work, our work is the first that proposes to learn useful intrinsic rewards across multiple lifetimes and across multiple tasks using a scalable meta-gradient method. More importantly, we believe that our empirical findings about interesting kinds of knowledge captured by intrinsic rewards (e.g., long-term exploration based on curiosity) and how they generalise to unseen agent-environment interfaces are new and worth more discussion, as intrinsic rewards have been receiving a lot of attention in recent years.

---

### Author Response · Authors · 2019-11-14
**Thank you very much for your additional additional comments.**

We have incorporated empirical work on ICM as suggested by R2 and discussed below. Since we have not yet seen engagement from R1 in the rebuttal process, any consideration on your part of (further) raising the score would be very much appreciated; this would allow the paper to be considered for acceptance if you would want to encourage that outcome. Thanks.

--------- Response to additional comments from R2 ---------
Thank you very much for the additional constructive comments.

# Regarding comparison to hand-designed intrinsic rewards on current tasks
We added ICM (Pathak’17) as an additional baseline (see Figure 3 and Section 4). Our method outperforms ICM on all four domains. We found that ICM does not explore effectively in our domains because the inverse model does not capture the uncertainty of the reward of each object, because actions can be predicted just from the agent’s movement.
Regarding comparison to Ostrovsky'17, both Bellemare’16 and Ostrovsky'17 are designed to approximate count-based exploration methods using a density model p(x). Thus, we believe that they the count-based exploration baseline in our paper, which uses the true state-visit counts, captures a comparison to those methods.

# Regarding comparison to approaches for changing action spaces
It would be indeed interesting to investigate deeper how the learned intrinsic rewards can be used to adapt to changing action spaces. We would like to leave it as future work given the time constraint. Thank you for suggesting an interesting future direction.

# Regarding how to use the learned intrinsic rewards
In this paper, we show several possibilities of using the learned intrinsic rewards, i.e., training new agents with different action spaces or new agents using different learning algorithms. More possibilities include being combined with policy transfer methods to further improve fast adaptation performance, generalising to unseen tasks, etc. We hope our work can inspire more research towards this direction.

--------- Response to additional comments from R3 ---------
Thank you very much for increasing the score. We address the questions below and incorporated some of the suggestions in the revision.

# Regarding degraded performance on Key-Box
We added the “near-optimal” curve on the key-box domain in Figure 3 - thank you for pointing this out. We found that the near-optimal performance is also worse compared to Random ABC due to the variance when sampling rewards for each object (30 randomly sampled environments are used during evaluation), though they should be the same in expectation. We noticed that some objects are sometimes not reachable due to the randomised locations, which may also explain the slight degradation. We observed that the agent has learned the same qualitative “explore-and-then-exploit” behaviour as well as navigating to the correct object.

---

> ### Comment · AnonReviewer2 · 2019-11-14
> **appreciate yet another experiment; authors performed a significant amount of extra work; support acceptance**
>
> With a brief check, the authors' explanation that the proposed method outperforms ICM due do uncertainty of the reward for each object seems to be a very convincing explanation as current hand-designed intrinsic rewards typically struggle in uncertain environments. Due to this year's changed scoring scale, I am unable to increase my score, but I believe the paper is now a solid accept and would be worth 7/10 by previous year scale. The authors have performed a significant amount of extra work and seem to have engaged constructively and promptly with all reviewers during the rebuttal period.

---

### Decision · Program_Chairs · 2019-12-19

**Decision:**

Reject

**Comment:**

The authors present a metalearning-based approach to learning intrinsic rewards that improve RL performance across distributions of problems.  This is essentially a more computationally efficient approach to approaches suggested by Singh (2009/10).  The reviewers agreed that the core idea was good, if a bit incremental, but were also concerned about the similarity to the Singh et al. work, the simplicity of the toy domains tested, and comparison to relevant methods.  The reviewers felt that the authors addressed their main concerns and significantly improved the paper; however the similarity to Singh et al. remains, and thus the concerns about incrementalism.   Thus, I recommend this paper for rejection at this time.